**Cite this article:** Östergren J, Palm S, Gilbey J, Spong G, Dannewitz J, Königsson H, Persson J, Vasemägi A. 2021 A century of genetic homogenization in Baltic salmon—evidence from archival DNA. *Proc. R. Soc. B* **288**: 20203147.

genetics, molecular biology

*Salmo salar*, historical DNA, human-induced, genetic change, conservation

**Author for correspondence:**
Johan Östergren
e-mail: johan.ostergren@slu.se

# A century of genetic homogenization in Baltic salmon—evidence from archival DNA

Johan Östergren[1], Stefan Palm[1], John Gilbey[2], Göran Spong[3], Johan Dannewitz[1], Helena Königsson[3], John Persson[1] and Anti Vasemägi[1,4]

[1]Swedish University of Agricultural Sciences, Department of Aquatic Resources, Institute of Freshwater Research, Stångholmsvägen 2, SE-178 93 Drottningholm, Sweden
[2]Marine Scotland Science, Freshwater Fisheries Laboratory, Faskally, Pitlochry, PH16 5LB, UK
[3]Department of Wildlife, Fish, and Environmental Studies, Swedish University of Agricultural Sciences, SE-901 83 USA
[4]Chair of Aquaculture, Institute of Veterinary Medicine and Animal Sciences, Estonian University of Life Sciences, Tartu, Estonia

JÖ, 0000-0002-7585-7629; SP, 0000-0002-9890-8265; JG, 0000-0002-5064-0589; GS, 0000-0002-1246-5046; JD, 0000-0003-3548-6023; AV, 0000-0002-2184-5534

Intra-species genetic homogenization arising from anthropogenic impacts is a major threat to biodiversity. However, few taxa have sufficient historical material to systematically quantify long-term genetic changes. Using archival DNA collected over approximately 100 years, we assessed spatio-temporal genetic change in Atlantic salmon populations across the Baltic Sea, an area heavily impacted by hydropower exploitation and associated with large-scale mitigation stocking. Analysis was carried out by screening 82 SNPs in 1680 individuals from 13 Swedish rivers. We found an overall decrease in genetic divergence and diminished isolation by distance among populations, strongly indicating genetic homogenization over the past century. We further observed an increase in genetic diversity within populations consistent with increased gene flow. The temporal genetic change was lower in larger wild populations than in smaller wild and hatchery-reared ones, indicating that larger populations have been able to support a high number of native spawners in relation to immigrants. Our results demonstrate that stocking practices of salmon in the Baltic Sea have led to the homogenization of populations over the last century, potentially compromising their ability to adapt to environmental change. Stocking of reared fish is common worldwide, and our study is a cautionary example of the potentially long-term negative effects of such activities.

## 1. Introduction

Human-driven loss of biodiversity at several levels is threatening the functioning of ecosystems at a global scale [1,2] as we now live in the 'sixth extinction' geological epoch [3]. In addition to extinction, the loss of biodiversity may occur even without drastic or apparent changes in population abundance. For example, inter- or intraspecific hybridization can lead to the breakdown of reproductive barriers and previously distinctive populations or species boundaries, a process known as genetic homogenization (GH). GH is extremely widespread and increasingly recognized as a common cause of diversity loss [4]. Breakdown of historical genetic population structures, leading to potentially locally adapted subgroups of a species becoming genetically admixed, may potentially erase the outcomes of long-term evolutionary processes having shaped specific adaptations [5]. It further may reduce meta-population resilience arising through mechanisms such as the 'portfolio effect' [6,7].

Intraspecific genetic diversity not only maintains environmental adaptations [8], it also defines a species's adaptive potential; evolution in a heterogeneous environment produces and maintains adaptive genetic variation which in turn can be a major determinant of the capacity of populations to adapt to changing environmental conditions [9]. The ability to adapt to such environmental change is reliant on the inherent genetic variation both across [6,7] and within populations [10], and the loss of such variation may lead to long-term risks to population viability [11]. Inbreeding leads to reductions in intraspecific genetic diversity and is in that way linked to population fitness. Outbreeding increases genetic diversity, but may lead to the breakdown of local adaptation potentially resulting in associated decreases in fitness [12,13].

Studying genetic change over longer time scales is a key aspect of conservation and sustainable management [14]. Access to historical DNA in this context provides unique opportunities to study genetic processes over time, and several studies have used archival material to investigate temporal genetic change in mammals [15,16], birds [17] and fishes [18–20]. However, studies have often been limited by the small number of available historical samples or by restricted geographical and/or temporal coverage.

Releases of hatchery-reared fish to compensate for habitat loss and/or to increase fishing opportunities are common worldwide. While in some cases such interventions may be beneficial for assisting threatened populations, a number of genetic conservation concerns have been identified (fitness reductions, reduced effective population size, reduced fitness of recipient populations, etc.) that depend on how artificial propagation and stocking is carried out [11,21–24]. Several studies have also documented GH of native populations due to large-scale interbreeding with hatchery strains, including decreases in genetic structuring and isolation by distance (IBD) relationships [25–29]. By contrast, there are also situations where only limited genetic introgression of released non-native strains into wild populations have been documented, despite extensive hatchery interventions [30,31]. Thus, it is of paramount importance that stocking programs are monitored to ensure no negative impacts are realized (e.g. [32,33]).

The Atlantic salmon (*Salmo salar* L.) is a keystone species with high historically and contemporarily social, economic and cultural value. During the last century, Atlantic salmon populations have suffered from a range of anthropogenic activities such as hydropower development, habitat degradation and overfishing. Salmon in the Baltic Sea are no exception [34]. Historically, at least 80 rivers flowing to the Baltic Sea supported wild salmon populations, but today only 28 remain [35]. Sixteen of the rivers with wild populations are situated in Sweden, providing the Baltic Sea with approximately 90% of wild salmon juveniles (smolts) [35]. To compensate for production losses in rivers exploited by hydropower and other anthropogenic effects, annual releases in several countries of in total approximately 5 million hatchery-reared smolts are undertaken in the Baltic Sea (approx. 1.7 million in Sweden), which accounts for around 60% of the total (wild + reared) annual smolt production [35]. The share of reared smolts was even higher (approx. 90%) in the 1980–1990s when the status of wild populations was alarmingly low (electronic supplementary material, figure S1). Given these circumstances, the Baltic salmon history during the last century represents one of the most extensive and long-term 'stocking experiments' in the world. To date, however, no comprehensive genetic study has addressed long-term effects of these large-scale sea-ranching and stocking activities that have been ongoing for more than 60 years [36].

In this study, we took advantage of a unique archived scale collection from Baltic salmon, collected by fishermen and fishery biologists in Sweden from the 1920s onwards. This collection provides a unique opportunity to describe long-term temporal genetic changes, and to relate contemporary findings to historical records of stocking and other anthropogenic activities. We characterized genetic changes over the last 100 years by studying archived scales from 13 Swedish Baltic salmon populations across the 1400 km latitudinal gradient, collected before and during the emergence of major anthropogenic pressures. In particular, we aimed to (i) compare the historical (1920s–1930s) genetic population structure with the contemporary state (2010–2015), (ii) quantify the amount and direction of genetic change between and within populations, and (iii) assess the relative role of different evolutionary forces that have led to the present genetic structure. In addition, we compared changes in wild and hatchery-reared populations, as a means to disentangle natural and anthropogenic processes behind observed temporal changes.

## 2. Material and methods

### (a) Sample collection

A total of 956 historical and 814 contemporary individual samples from Baltic salmon in 13 Swedish rivers were analysed. The salmon rivers studied are located along with a coastline distance of greater than 1400 km. These span from large northern rivers (e.g. Kalix and Torneälven) with a yearly spawning run of 10 000–150 000 adults to much smaller rivers (e.g. Öreälven and Rickleån) with only 50–200 spawning adults. Seven large rivers (e.g. Lule and Umeälven) are completely inaccessible due to dams for hydropower generation and today have no wild salmon production. (figure 1; electronic supplementary material, table S1). Five of the rivers studied are today heavily regulated for hydropower production; since the 1950s salmon populations in those rivers have consisted solely of hatchery-reared (sea-ranched) salmon released as compensation for lost fishing opportunities.

Our dataset thus comprises eight rivers that still have natural salmon production (hereafter termed 'wild') and five rivers where salmon production is fully supported by hatcheries (hereafter termed 'reared') (electronic supplementary material, table S1). At present, no hatchery stocking is undertaken in any of the wild populations. However, there has been historic stocking in all of the wild rivers, mainly with locally caught salmon (i.e. supportive breeding), but included the occasional use of non-native salmon.

For comparative reasons, we pre-defined the following groupings when analysing our data: (A) all populations, (B) reared populations, (C) wild populations, and (D) wild populations except the southernmost one (Mörrumsån) that is located very distant from all other studied rivers (figure 1). The last group was considered since the Mörrumsån salmon population is genetically distinct compared to more northern ones in the Baltic Sea, possibly reflecting postglacial colonization from a separate refugia [37].

Historical samples were collected from 1920s–1930s (*n* = 11 populations), 1946 (one population) and 1961–1963 (one population) while the contemporary samples originated from 2010–2015 (electronic supplementary material, table S1). Historical

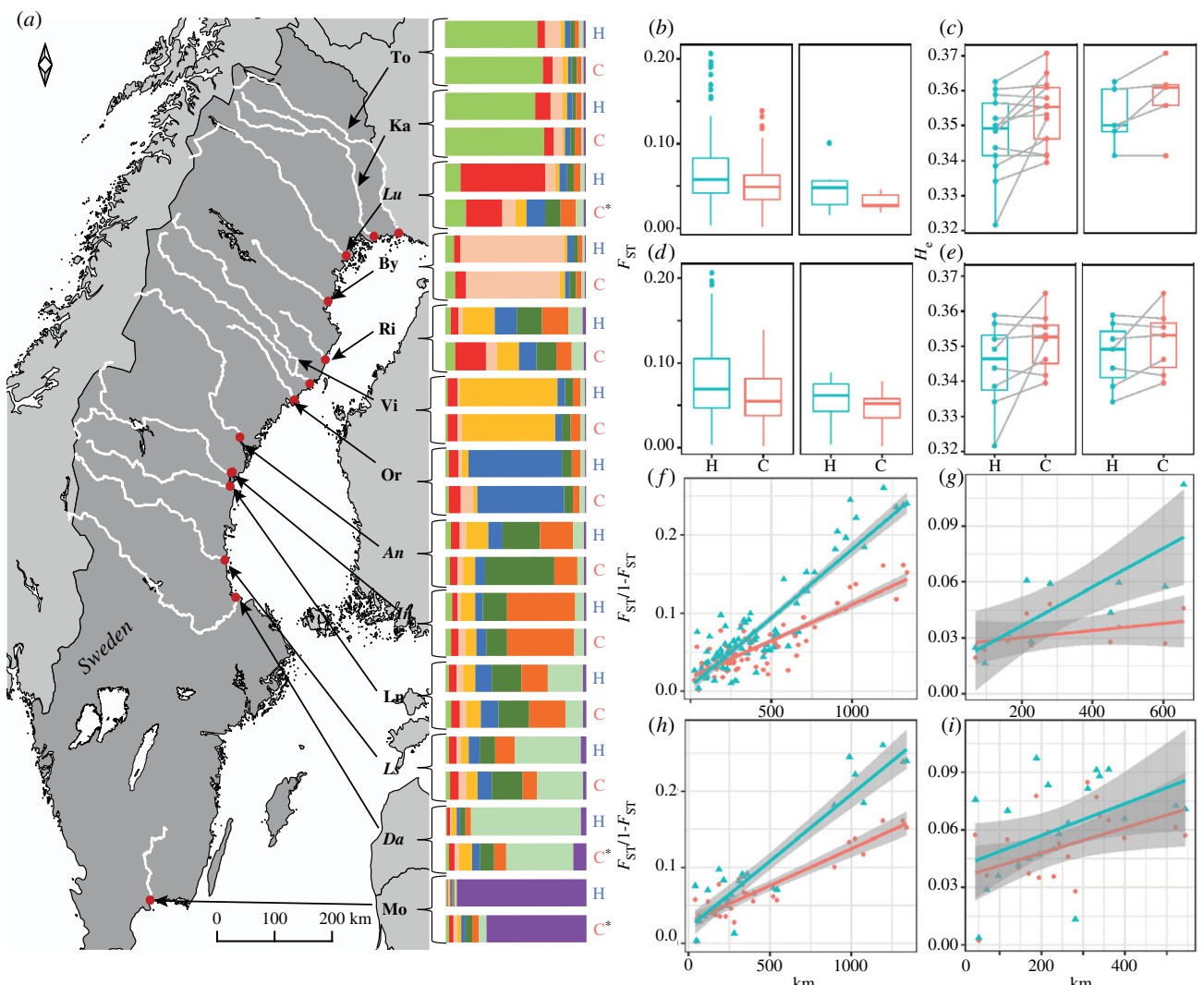

**Figure 1.** Clustering analysis results from Structure ($K = 9$) of temporal samples (H = historical, C = contemporary) from 13 salmon populations in the Baltic Sea. Each cluster is represented by a colour showing summaries of cluster proportions. * indicates significant change in historically dominant cluster proportion. River/population abbreviations same as in electronic supplementary material, table S1. Text indicates current status (normal = wild, italic = reared) (right of map (a) in figure). Average pairwise $F_{ST}$ and diversity $H_e$ are shown per pre-defined population groups ($b,c$ = all and reared; $d,e$ = wild and wild without Mörrum). Isolation by distance (IBD) as genetic ($F_{ST}/1\text{-}F_{ST}$) versus geographic distance (closest waterway distance in km) per pre-defined population group showing 95% confidence regions ($f$ = all, $g$ = reared, $h$ = wild, $i$ = wild without Mörrum). Colours indicate temporal time period (blue = historical, orange = contemporary). (Online version in colour.)

samples consisted of dried scales from adult fish stored in labelled paper envelopes at the Swedish University of Agricultural Sciences (Älvkarleby and Drottningholm). Contemporary samples (fin tissue stored in 95% ethanol) were collected in rivers, either from juveniles (parr and smolt) or adult spawners.

## (b) SNP panel development

The genetic markers employed were a subset of those described by Karlsson *et al.* [38]. Because these markers were developed for Atlantic salmon in Norway (genetically distinct from Baltic populations), we pre-screened altogether 160 SNP markers for their technical and biological performance in Swedish Baltic salmon samples. First, only markers that showed well-defined clustering patterns in the genotyping step on the Fluidigm Biomark EP1 platform were retained. Second, markers were evaluated based on their amplification rate, minor allele frequency (MAF), Hardy–Weinberg equilibrium (HWE) and distribution of alleles (electronic supplementary material, table S2). We excluded markers where: (i) the overall amplification rate was less than 85%; (ii) the minor allele frequency was less than 0.1; (iii) the allele distribution was deviated significantly from

expected ranges for HWE; or (iv) the heterozygote and both homozygotes for all autosomal markers (see below) were not present in our dataset. The final set of markers comprised 82 autosomal SNPs (electronic supplementary material).

## (c) Genetic analysis and quality control

DNA was extracted from scales using a Qiagen Symphony robot and the QiaSymphony DNA tissue kit according to the manufacturer's instructions. Extraction, PCR amplification and SNP genotyping all took place in separate access controlled rooms with dedicated equipment. Genotyping was performed on a Fluidigm Biomark microarray platform using the 96.96 dynamic array, following standard protocols with the exception of historic samples being pre-amplified for 30 cycles instead of the standard 14 used for the contemporary samples. All genotyping runs included three NTC (no template controls) and positive control reference samples. Samples were run in duplicates for quality control and error estimates. Only individuals that showed genotype completion greater than 95% were retained in the final analyses. Samples were checked for contamination by ensuring that they fell within the expected range for heterozygosity

levels (contaminated samples show heterozygote excess). To quantify the level of genotyping error caused by a lack of amplification of one of the alleles (dropout), this rate was calculated from samples run in duplicates. The true dropout rate is likely to have been somewhat higher, but since we cannot detect dropout in homozygotes, this could not be easily established.

### (d) Outlier SNP analysis

A series of tests were carried out aimed at detecting putative outliers among the 82 SNPs (i.e. loci showing higher or lower genetic differentiation than expected under assumed neutrality). Three different statistical approaches were employed using the software ARLEQUIN [39], BAYESCAN [40] and OUTFLANK [41]. Historical and contemporary samples were always analysed separately. The presence of hierarchical genetic structuring may yield spurious significances when scanning for outliers [42]. Therefore, analyses were run both including ($s = 13$) and excluding ($s = 12$) the genetically and geographically most deviating population (Mörrumsån). When that deviating sample was included, both a non-hierarchical and a hierarchical analysis was performed with ARLEQUIN. In total, 34 separate tests were carried out using different combinations of samples and statistical methods. Full details of all outlier analyses and program settings are described in electronic supplementary material, appendix S1 .

### (e) Statistical treatment of data

To counteract potential bias on genetic parameters due to non-representative sampling caused by family structure [43–46] we screened the data for the existence of full siblings using the maximum-likelihood approach implemented in COLONY 2.0.4.4 [47,48] We subsequently kept a maximum of two randomly selected individuals from each inferred full-sibling family.

We used the program STRUCTURE [49] to investigate genetic population structure in our historical and contemporary samples with emphasis on potential temporal change. STRUCTURE uses individual genotypic data to infer the number of populations or genetic clusters ($K$) that give the best/optimal solution based on the assumption that clusters should be in Hardy–Weinberg and linkage equilibrium. Individuals can be assigned to one or, in case of admixed ancestry, several clusters. We used a model assuming admixture with correlated allele frequencies [50] and a burn-in of 100 000 and run length of 100 000 Markov chain Monte Carlo (MCMC) iterations. The number of clusters ranged from $K$ 5 to 15, with 10 replicated runs per $K$. To define the most likely number of clusters, we applied the $\Delta K$ method [51], as implemented in the software STRUCTURE HARVESTER [52].

Collections of individual genotypes that share clusters are presumed to have a closer genetic relationship to each other. The biological meaning of clusters identified by STRUCTURE (e.g. along genetic clines) has been discussed extensively [53–55]. In our study, we used the cluster information for spatial and temporal comparisons. We grouped all samples (i.e. historical and contemporary) in a single STRUCTURE analysis and compared cluster distributions over time. Differences in such temporal genetic changes (proportions of membership in dominant cluster $K$) across samples (populations) were investigated with the prop. test function in R [56]. Historical and contemporary samples were also analysed separately for comparative reasons.

The program FSTAT [57] v. 2.9.3.2 was used to estimate expected heterozygosity $H_e$, $F_{IS}$, global and pairwise $F_{ST}$ [58]. The same program was also used to conduct tests for genetic differentiation ($F_{ST}$) between pairs of samples (6500 permutations), between pre-defined population groups (1000 permutations), and for deviations from Hardy–Weinberg equilibrium (42 640 randomizations). Differences in pairwise $F_{ST}$ and $H_e$ were investigated with permutation tests in FSTAT and in R with Wilcoxon signed-rank paired test. We also performed an

analysis of power to detect genetic heterogeneity for the present 82 SNP markers (full details given in electronic supplementary material, appendix SA2). The relationship between genetic ($F_{ST}/1-F_{ST}$) and geographic distance (pairwise waterway in km) was investigated using a Mantel test and linear regression in R. We tested for differences in regression intercepts and slopes using the R packages emmeans and lsmeans::lstrends. Pairwise waterway distances were calculated using a distance matrix for a number of points on a uniform raster, using the R package gdistance and its function costDistance. costDistance calculates the shortest path between points on a raster where raster values represent landscapes 'friction'. In our case, all cells with water had value 1 whereas all cells including land were set as NA; this yielded the shortest path between points (river estuaries) without crossing land.

A multidimensional scaling analysis (MDS) based on 82 SNP genotypes was performed using the adegenet package in R [59]. A dendrogram (unrooted, neighbour-joining) based on pairwise chord distances [60] was created using PHYLIP [61] and visualized with FIGTREE V. 1.4 [62]. Bootstrap support values for branches were created based on 1000 replications.

### (f) Evaluating the relative importance of drift and gene flow

We produced population-specific estimates of average immigration rate ($m$) and local variance effective size per generation ($N_e$), following the method described by Wang & Whitlock [63] that assumes immigration from an infinitely large (temporally stable) source population into a focal population. Specifically, we used the maximum-likelihood estimator implemented in software MLNE [63] and analysed each temporal sample pair against the same large pooled sample encompassing all historical and contemporary data except Mörrumsån (natural gene flow from this southern river into northern ones is likely very limited, and no such stock transfers have been documented). Based on latitudinal differences in average smolt age, average generation intervals (needed when estimating $N_e$ and $m$) were varied from $G = 5$ years in the south (Mörrumsån) to $G = 7$ in the north (Torne and Kalixälven), with $G = 6$ for the 10 intermediate rivers. We similarly adjusted temporal time intervals between our focal samples to make them compatible between rivers, by accounting for if adults, smolts and/or parr had been analysed (see electronic supplementary material, table S1).

## 3. Results

Of the total number of individual samples analysed, 5.5% (historical) and 1.7% (contemporary) were excluded due to an incomplete genotype (i.e. less than 95% loci genotyped). An additional 10 historical and 13 contemporary individuals were excluded following analysis with COLONY (full-sibling reduction). The final dataset thus comprised 893 historical and 787 contemporary individuals (electronic supplementary material, table S1). The overall genotyping error (drop out) rate based on samples run in duplicates was found to be well below 1% and thus unlikely to affect any of the inferences or conclusions in this study.

Outcomes of the various outlier tests carried out proved to be very sensitive to the parameter settings (electronic supplementary material, appendix SA1). Over all tests, a single locus (out of 82) appeared as potentially affected by directional selection under some test conditions. However, this locus was only identified as a putative outlier when historic or contemporary samples from Mörrumsån were included, and the

**Table 1.** Average pairwise $F_{ST}$ and average expected heterozygosity $H_e$ of pre-defined groups of populations: all (*a*), reared (*b*), wild (*c*) and wild without Mörrumsån (*d*). Values for historical and contemporary samples with 95% confidence interval (CI) are shown.

| population group | average pairwise $F_{ST}$ | | average expected heterozygosity $H_e$ | |
|---|---|---|---|---|
| | historical (95% CI) | contemporary (95% CI) | historical (95% CI) | contemporary (95% CI) |
| (A) | 0.074 (0.063–0.087) | 0.059 (0.051–0.067) | 0.347 (0.341–0.354) | 0.354 (0.349–0.359) |
| (B) | 0.046 (0.036–0.057) | 0.031 (0.026–0.037) | 0.353 (0.345–0.359) | 0.358 (0.350–0.367) |
| (C) | 0.101 (0.084–0.122) | 0.076 (0.065–0.088) | 0.344 (0.336–0.352) | 0.351 (0.346–0.360) |
| (D) | 0.054 (0.047–0.062) | 0.049 (0.040–0.058) | 0.348 (0.341–0.354) | 0.351 (0.345–0.358) |

outlier status of this SNP was not consistent across multiple genome scan approaches. In view of these observations, which suggested no strong and consistent selection at any loci, we retained all 82 SNPs for downstream statistical analyses.

Across all 82 SNPs, a deviation from HW equilibrium was evident in five of 26 samples, as $F_{IS}$ values were significantly different from zero (three in negative and two in positive direction, electronic supplementary material, table S3). However, only one sample, the historical sample from Luleälven (Lu_H), remained significant after correcting for multiple tests. Thus, we retained all 1680 individual samples for downstream statistical analyses.

The model-based clustering analysis using Structure [49] provided the highest support for nine genetic clusters ($K = 9$) based on the method described by Evanno *et al.* [51] for all samples analysed together and for historic samples analysed separately. These nine clusters generally corresponded well with the rivers sampled (figure 1; electronic supplementary material, figure S3). In 10 out of 13 rivers, a single cluster dominated (i.e. occurred at greater than 46% frequency) in the historical sample, whereas in three cases (Rickleån (Ri), Ångermanälven (An) and Ljungan (Ln)), no clear dominance of a cluster could be identified (figure 1). For two pairs of neighbouring rivers, the same cluster dominated; Torneälven (To) and Kalixälven (Ka), and Dalälven (Da) and Ljusnan (Ls), respectively. There was a significant temporal change in the proportion of the historically dominating cluster in four of eleven rivers (prop.test, $p < 0.05$). Three of those rivers, Luleälven (Lu), Dalälven (Da) and Mörrumsån (Mo), showed a decrease in the proportion of the historically dominating cluster (figure 1; electronic supplementary material, figure S2). For contemporary samples only, the highest support was yielded for $K = 7$. For Luleälven (Lu), no dominating cluster was evident, and Ångermanälven (An) and Indalsälven (In) shared a dominating cluster (electronic supplementary material, figure S3).

Genetic diversity measured as expected heterozygosity ($H_e$) showed a temporal increase within all the pre-defined population groups, but this increase was only statistically significant when all samples were studied together (group A, paired Wilcoxon signed-rank test $p < 0.05$; figure 1 and table 1). On the individual river level, estimates of $H_e$ was higher in nine of 13 contemporary samples compared to the corresponding historical sample. Three of four samples with lower contemporary than historical $H_e$ estimates were wild salmon populations (Torneälven (To), Rickleån (Ri) and Ljungan (Ln)) (electronic supplementary material, table S3).

Contemporary samples showed lower population divergence compared to historical samples. A significant decrease in average pairwise $F_{ST}$ (historic versus contemporary) was noted when comparing all populations (A) and wild populations only (C) (paired Wilcoxon signed-rank test $p < 0.05$; figure 1 and table 1). A trend towards genetic homogenization, albeit non-significant, was also observed for reared (B) and wild populations without Mörrumsån (D). Pairwise $F_{ST}$ across loci between historical and contemporary samples within populations varied between 0.001 and 0.030, and nine of the 13 comparisons exhibited a significant temporal genetic change (electronic supplementary material, table S4). By contrast, three large wild populations (rivers Kalixälven (Ka), Torneälven (To) and Vindelälven (Vi)) did not show any signs of genetic change. In line with pairwise allele frequency comparisons, global $F_{ST}$ across populations and loci decreased over time in all pre-defined groups (A–D), although non-significantly (electronic supplementary material, table S5). Power analysis showed that regardless of the number of samples and statistical method, power was 100% at $F_{ST} = 0.015$ (and above). When comparing two samples mimicking the present temporal comparisons, power was 80% at $F_{ST}$ greater than 0.005 (for Chi2) and at $F_{ST}$ greater than 0.009 (for Fisher's method) (electronic supplementary material, appendix SA2).

Both historical and contemporary samples exhibited significant isolation by distance (IBD) patterns for all rivers (A) and for only wild ones (C and D) (Mantel test, $p < 0.05$; linear regression, $p < 0.05$; figure 1). Among the currently reared populations, historical but not contemporary samples showed significant IBD. In addition, historical samples exhibited steeper relationships between geographic and genetic distances in all cases except for wild populations without Mörrumsån (D) (figure 1). However, even for the latter dataset, 17 out of 21 pairwise differences between wild populations showed reduced divergence among contemporary samples.

MDS based on a Neís genetic distance $D_A$ matrix (figure 2) and a neighbour-joining dendrogram (unrooted, chord distance) (electronic supplementary material, figure S4) gave further support to the findings based on $F_{ST}$ estimates, Bayesian clustering (Structure) and Mantel tests, thus underlining the overall reduction in population divergence in contemporary samples compared to historic ones (figure 2; electronic supplementary material, figure S4 and figure S5).

The analysis with Mlne yielded effective population ($N_e$) point estimates in the range *ca.* 50 to 1000, with values corresponding to relative differences in river catchment size (electronic supplementary material, table S1) and estimated potential productivity [35] except for Ljungan (Ln), for

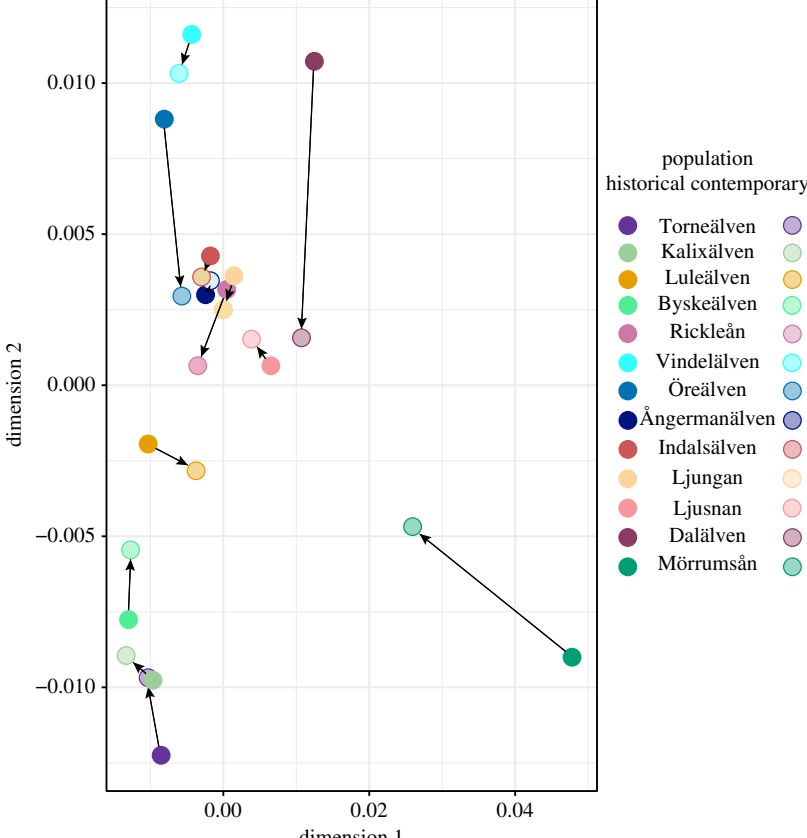

**Figure 2.** A multidimensional scaling analysis (MDS) based on Neís genetic distance $D_A$ matrix, illustrating two-dimensional structures of all samples. Black arrows indicate change from historical to contemporary samples. (Online version in colour.)

which $N_e$ was surprisingly high (figure 3). Estimated average immigration rates ($m$) varied in line with the STRUCTURE results. For larger wild populations, estimated immigration rates were comparably low (Torneälven: $m = 0.0149$, Kalixälven: $m = 0.0046$) while for smaller wild and reared populations $m$ was considerably higher (especially for Luleälven: $m = 0.0817$, and Dalälven: $m = 0.0638$; figure 3; electronic supplementary material, table S6). Furthermore, one small wild population, Rickleån, experienced an exceptionally high level of estimated immigration ($m = 0.2452$) compared to all other rivers (figure 3).

## 4. Discussion

The temporal genetic analysis spanning a century enabled us to obtain a comprehensive view of the impact of human-driven genetic changes arising from one of the oldest and most extensive stocking experiments in the world. Based on comparisons between archival DNA and contemporary samples from rivers on a geographical distance of over 1400 km, we provide strong evidence of genetic homogenization among Atlantic salmon in the Baltic Sea over the last 100 years. The observed temporal genetic changes, such as decreased pairwise $F_{ST}$ estimates and weakened IBD patterns, strongly indicate that human activities have led to elevated gene flow during the last century. Reduced genetic divergence between and increased diversity within populations further suggest that the changes are most likely to be connected to increased immigration, and that these changes are mainly driven by hatchery and stocking practices. Furthermore, joint maximum-likelihood analysis of gene flow and

effective population size revealed a pattern of genetic resilience of larger wild salmon populations which have been able to resist genetic change by supporting a large number of native spawners. By contrast, smaller wild or hatchery-reared populations have experienced marked genetic change.

A main driver behind the observed genetic homogenization is most likely anthropogenic activities coupled to hydropower exploitation. The majority (11 of 13) of our historical samples was collected in the 1920–1930s before large-scale development of hydropower production was initiated in the Baltic Sea area. In five of the studied rivers, this development peaked in the 1950s, which led to a diminishing wild salmon production that ended completely in the 1960–1970s [34,64]. To maintain salmon production and compensate fisheries, hatcheries were built and reared stocks replaced the natural populations. Since the 1960s, the number of released reared salmon in the Baltic Sea has been roughly 5 million annually, which was (and still is) considerably more than the wild smolt production [35]. In the beginning of the establishment of hatchery rearing (using native salmon as broodstock), certain problems such as disease outbreaks led to low hatchery production periodically, and hydropower companies thus used non-native salmon (mostly from other Swedish Baltic rivers) in all hatcheries to achieve obligatory amounts of released salmon set in court decisions. In most cases, the numbers of stocked non-native salmon were limited, with the exception of the river Luleälven, where salmon from more than 30 different populations of non-native origin have been released in large numbers over time, mostly during the 1960s [64]. Luleälven was also the population that showed the largest change in genetic composition over this period.

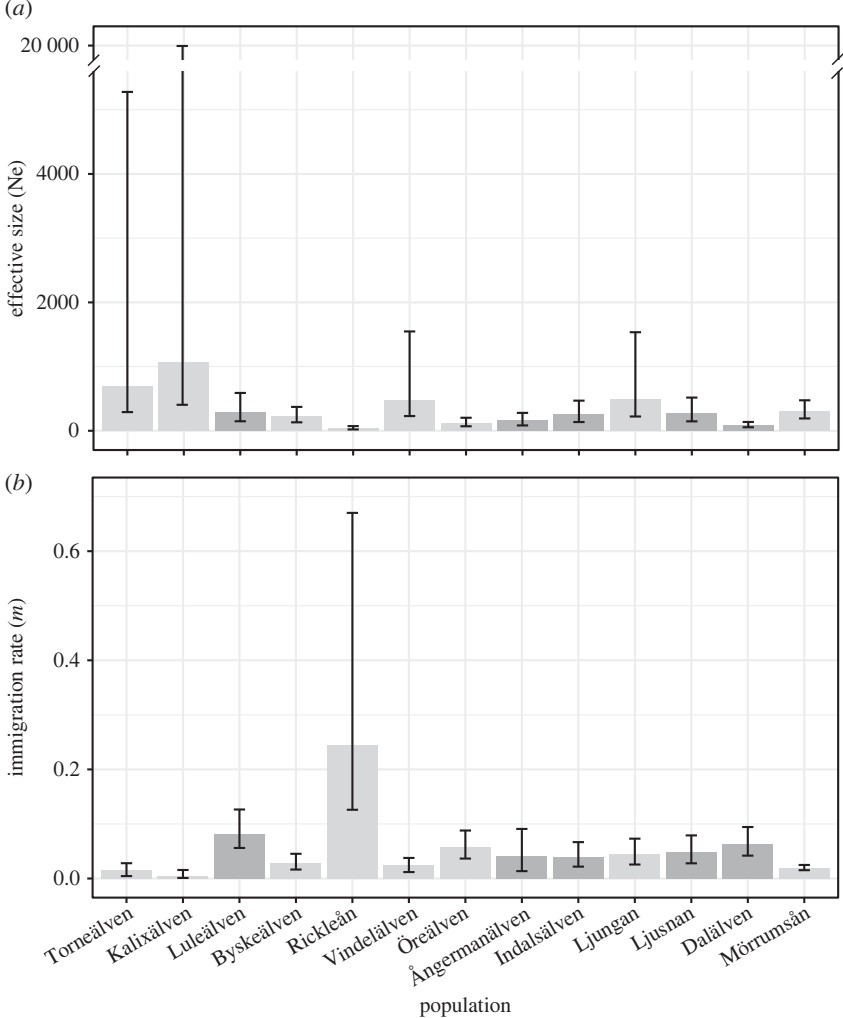

**Figure 3.** Results from the program MLNE showing estimated effective population size ($N_e$) (*a*) and (im)migration rate (*m*) (*b*). Current status as wild (grey) and reared (dark grey) is indicated.

Stocking of young salmon (eggs, fry and parr) directly into wild rivers has also been performed in the Baltic Sea area during the last century, mostly in limited numbers (electronic supplementary material, figure S1). In some wild rivers, however, larger amounts of young salmon were released for assisting populations threatened by a combination of habitat destruction, health issues and a high fishing pressure. As an example, the wild salmon population in river Rickleån, which has shown remarkable genetic change over time, was considered depleted due to anthropogenic activities (pulp industry) in the 1950s. Substantial stocking efforts using salmon from several non-native populations were thus early initiated in this small river.

In addition to mixing and releasing populations with different genetic backgrounds in non-native rivers, as applied periodically in some hatcheries historically, it is likely that the temporal decrease of genetic divergence seen among wild salmon populations reflects increased gene flow caused by elevated straying rates of released hatchery-reared fish [28,29]. There is also evidence from tag-recapture data that Baltic salmon have been widely spread following previous so-called *delayed release* (i.e. fish stocked in the sea rather than in rivers to increase their survival rate [65]) aimed at supporting salmon sea fisheries. Delayed release prevents the juveniles from imprinting on their natal rivers [66], drastically increasing straying rates.

Among the studied wild populations, river Mörrumsån provides one example where elevated straying potentially could explain the observed temporal genetic change. There have been no documented releases with non-native salmon juveniles in this river. However, Mörrumsån flows into the southern Baltic Sea, and it is thus located very close to the main feeding area for most Baltic salmon populations [35,67], which may impose an increased risk for straying. Notably, repeated large-scaled delayed release experiments with salmon from Finnish river Iijoki (northern Gulf of Bothnia) were also carried out in the mid-1990s at the Island of Bornholm, just some 100 km from Mörrumsån [65]. In addition, large-scaled releases of hatchery-reared smolts derived from adults collected in the lowermost part of Mörrumsån and so assumed to be of local origin took place during the 1980s and early 1990s, with an associated risk of including strayers as within the broodstock used. Thus, locally released hatchery-reared offspring from strayers from other rivers have likely spawned together with wild salmon, following their return to spawning grounds in the natal river.

During the last century, most wild Baltic salmon populations have experienced a sharp decline in abundance driven by habitat degradation [68] and hydropower development [34,35]. In addition, for many wild populations, very few spawners were recorded for some years in the 1990s

due to high fishing pressure in combination with high mortality due to the M74 syndrome [35]. Furthermore, the total number of spawners in the contemporary hatchery-reared populations (approx. 100–350 individuals in broodstock) has been estimated to be considerable higher in the wild before hydropower exploitation (greater than 1000; e.g. [64]). In a meta-population with largely isolated subpopulations such as Baltic salmon, reductions in local population abundance is expected to elevate the effect of genetic drift resulting in increased genetic divergence over time [5]. By contrast, we observed slightly decreased population divergence and increased levels of local heterozygosity, indicating that past reductions in local population sizes have been counteracted by elevated gene flow due to genetic mixing in hatcheries, stocking with non-native fish and/or straying of released reared adults. Straying and subsequent admixture may also be consistent with the increase in expected heterozygosity [69]. Thus, our findings are in line with the results of Ozerov et al. [29], who reported an increase in diversity and reduction in divergence in Gulf of Finland Atlantic salmon populations over 16 years (1996–2012). However, our work represents the first case where spatio-temporal genetic changes have been studied using archived material prior to hydropower development and subsequent large-scale anthropogenic impacts on Baltic salmon.

More generally, our findings are well in line with earlier studies reporting temporal declines in genetic divergence among populations of Baltic salmon [28,70] and Atlantic salmon in France [71]. Declines in genetic divergence as a result of stocking have also been reported in other salmonids, such as sea trout (Salmo trutta L.) [72], coho salmon (Onchorynchus kisutch) [73] and brook charr (Salvelinus fontinalis) [25]. As observed here, temporal changes in IBD patterns have also been documented for brook charr [25] and Californian steelhead (Oncorhynchus mykiss) [26]. The latter study reported a complete breakdown of a previous IBD pattern, suggesting that damming and genetic introgression from released hatchery fish had acted as main factors responsible for the observed genetic change.

In summary, this study has revealed that large-scale hatchery and stocking practices have resulted in a partial breakdown of the genetic population structure of Baltic salmon during the past century, threatening the integrity of both wild and reared populations. As expected, we observed higher genetic resilience of large wild salmon populations which have been able to withstand genetic change by supporting a large number of native spawners in relation to immigrants. By contrast, most smaller wild populations and reared stocks experienced more severe genetic changes as a result of increased immigration stemming from a combination of genetic drift, past stock transfers and releases, and elevated straying by hatchery-reared salmon. As such, this work is among the first to document large-scale and long-term trends of human-induced genetic homogenization in salmonid fishes.

Although the current results are generally not detailed enough to assess the relative importance of specific mechanisms associated with increased gene flow that have acted on specific populations (i.e. past stock transfers among hatcheries, releases and elevated straying from hatchery-reared salmon into wild rivers), our findings are important. Genetic homogenization reduces one important aspect of intraspecific diversity. It further compromises individual and population fitness by disrupting local adaptations, at least in short to medium term, through the replacement of locally adapted alleles with non-adaptive ones, which is expected to reduce resilience to future environmental changes [11,21,24]. Thus, for salmon populations in the Baltic Sea, genetic homogenization has likely resulted in negative biological consequences.

A fundamental understanding of the mechanisms affecting levels of genetic diversity and genetic divergence between historical, remnant wild and hatchery populations is crucial for the establishment of effective conservation and restoration strategies. Our study demonstrates the dangers of human-invented mitigation strategies of rearing and releasing fish in nature, highlighting the importance of long-term monitoring and evaluation of human activities. Stocking of reared fish is common worldwide and our study should serve as a cautionary example of the potentially long-term negative effects of such activities, potentially hampering conservation and restoration of fish populations.

**Ethics.** All sample collection complied with applicable national legislation (Sweden).

**Data accessibility.** The data are available from the Dryad Digital Repository: https://doi.org/10.5061/dryad.2bvq83bpb [74].

**Authors' contributions.** J.Ö., S.P., J.D., G.S., J.G. and A.V. participated in the design of the study and drafted the manuscript; J.Ö. coordinated the study; H.K. carried out the molecular laboratory work; H.K. and G.S. carried out molecular data analysis; J.Ö., S.P., J.D., G.S., J.G. and A.V. carried out the statistical analyses and critically revised the manuscript; J.P. and J.Ö. collected field data and critically revised the manuscript; all authors gave final approval for publication.

**Competing interests.** The authors declare they have no conflicting interests with the work herein.

**Funding.** This study was funded by The Swedish Research Council Formas (Grant/Award 2013-1288).

**Acknowledgements.** We thank all fishermen, fishery biologists and hatchery staff that in the past and present have sampled salmon and arranged dried scale collections, allowing this study of unique historical samples. We further would like to thank our late colleague Tore Prestegaard for participating in the early phase of the project, and for his work in the genetic laboratory. Anders Kagervall is thanked for providing the map in figure 1 and for computing the distance matrix. This study was funded by the Swedish Research Council Formas (Grant/Award Number: 2013-1288).

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
