## [Peer Review File · Proceedings of the Royal Society B: Biological Sciences]

Review History

RSPB-2020-3147.R0 (Original submission)

Review form: Reviewer 1

Recommendation

Accept with minor revision (please list in comments)

Scientific importance: Is the manuscript an original and important contribution to its field?

Excellent

General interest: Is the paper of sufficient general interest?

Excellent

Quality of the paper: Is the overall quality of the paper suitable?

Excellent

Is the length of the paper justified?

Yes

Should the paper be seen by a specialist statistical reviewer?

No

Do you have any concerns about statistical analyses in this paper? If so, please specify them explicitly in your report.

No

It is a condition of publication that authors make their supporting data, code and materials available - either as supplementary material or hosted in an external repository. Please rate, if applicable, the supporting data on the following criteria.

Is it accessible?

Yes

Is it clear?

Yes

Is it adequate?

Yes

Do you have any ethical concerns with this paper?

No

Comments to the Author

This study examines archival and contemporary DNA collected over 100 years to assess spatio-temporal change in Atlantic salmon populations across the Baltic sea, an area highly impacted by hydropower exploitations and associated large-scale mitigation stocking. The authors find an overall decrease in genetic divergence and diminished isolation by distance among populations. Populations in rivers with hydropower and subsequent large-scale hatchery programs showed particularly large shifts in population structure over time including changes in relative proportions of genetic clusters and reduced population differentiation. The findings of this study are relevant to ongoing conservation and commercial concerns regarding Atlantic salmon. This study will also be of interest to researchers studying Pacific salmon species, many of which face similar challenges. This study is well-designed and well-written and I only have minor comments.

Specific comments:

Line 85: "Salmon in the Baltic Sea is" should be changed to "Salmon in the Baltic Sea are"

Line 90: Are the hatchery releases in Sweden or other countries?

Line 189-198: Were historic and contemporary samples examined separately with structure? It would be interesting to see if the historic samples showed even stronger clustering with contemporary samples removed from analysis.

Line 258: in what direction did the FIS of these sample differ? Elevated heterozygosity can be a sign of DNA contamination while very low (near zero) heterozygosity is an indicator that the wrong species was sampled. This last case happens somewhat often with pacific salmon but seems unlikely with Atlantic salmon.

Line 269-270: Can you indicate on figure 1 which populations had a significant change in the proportion of the historically dominating cluster? Maybe with an asterisk next to the cluster proportions?

Line 348: I like the discussion about the hatchery history on the river Luleälven. Do the other hatchery-dominated rivers have similar histories (stocking with non-native stocks) or did they use native fish to establish hatcheries? I believe this is important to note, even just briefly, either in discussion or methods because it could change the expected outcome in terms of population composition.

Line 360-379: I really like the discussion on the potential for elevated straying and how this could influence the observed changes in population structure. Straying and subsequent admixture may also be consistent with the increase in expected heterozygosity (Boca, Huang, and Rosenberg 2020)

Figure 1: It would help to have letter designations on the four top right panels, similar to the

bottom four right panels.

Figure 1. Are these true structure plots or summaries of the cluster proportion? I am used to seeing at least some individuals assigned to multiple clusters with structure but I don't see that here. It may just be that the size of the figures makes it difficult to see these individuals.

Figure 2: I like this figure. It's a nice way to show the different populations becoming more genetically similar over time.

Citations:

SM Boca, L Huang, NA Rosenberg (2020) On the heterozygosity of an admixed population. Journal of Mathematical Biology 81: 1217-1250

Review form: Reviewer 2

Recommendation

Accept with minor revision (please list in comments)

Scientific importance: Is the manuscript an original and important contribution to its field?

Good

General interest: Is the paper of sufficient general interest?

Good

Quality of the paper: Is the overall quality of the paper suitable?

Good

Is the length of the paper justified?

Yes

Should the paper be seen by a specialist statistical reviewer?

No

Do you have any concerns about statistical analyses in this paper? If so, please specify them explicitly in your report.

No

It is a condition of publication that authors make their supporting data, code and materials available - either as supplementary material or hosted in an external repository. Please rate, if applicable, the supporting data on the following criteria.

Is it accessible?

N/A

Is it clear?

Yes

Is it adequate?

Yes

Do you have any ethical concerns with this paper?

No

Comments to the Author

This paper presents a population genetic analysis of modern and historical samples of Atlantic salmon from multiple populations and demonstrates changes in diversity and divergence over

time. The authors attribute this change to hatchery stocking and other anthropogenic impacts. It is well done and the analysis appears sound. I found the structure of the presentation to be clear and logical. While the data consist of a small number of SNP loci by today's standards (82), they appear to provide sufficient variation for the questions addressed, although there is no formal power analysis.

This paper has many interesting parallels with work on Pacific salmonid species (*Oncorhynchus*), and I encourage the authors to dig more deeply into that literature for interesting comparisons in response to stocking (and, as someone who works mostly on those species, I commit to paying more attention to papers on *Salmo* species.)

Line 62; loss of genetic diversity in itself does not lead to inbreeding, and outbreeding would increase genetic diversity. Suggest editing this sentence for clarity.

Line 78; I think it would be clearer to add the word 'only', ie '...situation where only limited introgression...'

125-126; Can you provide any information on how much hatchery stocking actually occurs in the 'wild' populations? Or are they purely natural spawning systems?

Results

Line 266; I understand what you mean by 'dominated', but it would be good to define. Occurred at >50% frequency? Or whatever criteria you want.

Line 274; Did you consider other measures of genetic diversity besides heterozygosity? For microsatellites, allelic richness is a good measure, but for SNPs it can be informative to look at the proportion of polymorphic loci. This is just a suggestion, heterozygosity works.

Line 283; "A significant divergence in average pairwise F_{st} ..." Do you mean 'decrease' instead of divergence?

Lin 309; '...corresponding fairly well...' Perhaps reword, as Figure 3 doesn't provide any indication of the correlation between estimate N_e and river size or potential population size.

320; the novelty claim 'for the first time' is inaccurate for such a global statement; it may be true for this specific system, but such a claim seems unnecessary.

364-366; For non-salmonid biologists I think you should be clearer here about the implications of these releases directly into the ocean, i.e. that it prevents the juveniles from imprinting on their natal rivers, drastically increasing straying rates.

Interpretation of relative change in large vs small populations, e.g. line 413... author attribute it to effects of high immigration, but small populations will also experience greater changes from drift, confounding this interpretation.

Line 421; should be intraspecific diversity.

Line 423; I think this needs to be tempered; if locally adapted alleles truly have a selective advantage they will not be replaced by non-adaptive alleles (unless the amount of immigration is very high). Also I think you should delete the word 'give' from this sentence.

Best regards,
Devon Pearse

Decision letter (RSPB-2020-3147.R0)

28-Jan-2021

Dear Dr Ostergren:

Your manuscript has now been peer reviewed and the reviews have been assessed by an Associate Editor. The reviewers' comments (not including confidential comments to the Editor) and the comments from the Associate Editor are included at the end of this email for your reference. As you will see, the reviewers and the Editors have raised some concerns with your manuscript and we would like to invite you to revise your manuscript to address them.

Research ethics:

Use of animals and field studies:

It is a condition of publication that you make available the data and research materials supporting the results in the article. Please see our Data Sharing Policies (<https://royalsociety.org/journals/authors/author-guidelines/#data>). Datasets should be deposited in an appropriate publicly available repository and details of the associated accession number, link or DOI to the datasets must be included in the Data Accessibility section of the

article (<https://royalsociety.org/journals/ethics-policies/data-sharing-mining/>). Reference(s) to datasets should also be included in the reference list of the article with DOIs (where available).

Please submit a copy of your revised paper within three weeks. If we do not hear from you within this time your manuscript will be rejected. If you are unable to meet this deadline please let us know as soon as possible, as we may be able to grant a short extension.

Best wishes,
Professor Hans Heesterbeek
mailto: proceedingsb@royalsociety.org

Associate Editor
Board Member: 1
Comments to Author:

Both reviewers are very enthusiastic, and made just a few suggestions for how to improve the manuscript. Having read the paper myself, I also find the work of high quality.

Both reviewers make several constructive suggestions that the authors should be able to easily address. In particular, Reviewer 2 suggests a formal power analysis to determine whether the variation observed is sufficient for the question. Reviewer 1 asks whether the patterns observed are also evidence if you cluster the historic and contemporary samples separately.

Reviewer(s)' Comments to Author:

Referee: 1

Comments to the Author(s)

This study examines archival and contemporary DNA collected over 100 years to assess spatio-temporal change in Atlantic salmon populations across the Baltic sea, an area highly impacted by hydropower exploitations and associated large-scale mitigation stocking. The authors find an overall decrease in genetic divergence and diminished isolation by distance among populations.

Populations in rivers with hydropower and subsequent large-scale hatchery programs showed particularly large shifts in population structure over time including changes in relative proportions of genetic clusters and reduced population differentiation. The findings of this study are relevant to ongoing conservation and commercial concerns regarding Atlantic salmon. This study will also be of interest to researchers studying Pacific salmon species, many of which face similar challenges. This study is well-designed and well-written and I only have minor comments.

Specific comments:

Line 85: "Salmon in the Baltic Sea is" should be changed to "Salmon in the Baltic Sea are"

Line 90: Are the hatchery releases in Sweden or other countries?

Line 189-198: Were historic and contemporary samples examined separately with structure? It would be interesting to see if the historic samples showed even stronger clustering with contemporary samples removed from analysis.

Line 258: in what direction did the FIS of these sample differ? Elevated heterozygosity can be a sign of DNA contamination while very low (near zero) heterozygosity is an indicator that the wrong species was sampled. This last case happens somewhat often with pacific salmon but seems unlikely with Atlantic salmon.

Line 269-270: Can you indicate on figure 1 which populations had a significant change in the proportion of the historically dominating cluster? Maybe with an asterisk next to the cluster proportions?

Line 348: I like the discussion about the hatchery history on the river Luleälven. Do the other hatchery-dominated rivers have similar histories (stocking with non-native stocks) or did they use native fish to establish hatcheries? I believe this is important to note, even just briefly, either in discussion or methods because it could change the expected outcome in terms of population composition.

Line 360-379: I really like the discussion on the potential for elevated straying and how this could influence the observed changes in population structure. Straying and subsequent admixture may also be consistent with the increase in expected heterozygosity (Boca, Huang, and Rosenberg 2020)

Figure 1: It would help to have letter designations on the four top right panels, similar to the bottom four right panels.

Figure 1. Are these true structure plots or summaries of the cluster proportion? I am used to seeing at least some individuals assigned to multiple clusters with structure but I don't see that here. It may just be that the size of the figures makes it difficult to see these individuals.

Figure 2: I like this figure. It's a nice way to show the different populations becoming more genetically similar over time.

Citations:

SM Boca, L Huang, NA Rosenberg (2020) On the heterozygosity of an admixed population. *Journal of Mathematical Biology* 81: 1217-1250

Referee: 2

Comments to the Author(s)

This paper presents a population genetic analysis of modern and historical samples of Atlantic salmon from multiple populations and demonstrates changes in diversity and divergence over time. The authors attribute this change to hatchery stocking and other anthropogenic impacts. It is well done and the analysis appears sound. I found the structure of the presentation to be clear

and logical. While the data consist of a small number of SNP loci by today's standards (82), they appear to provide sufficient variation for the questions addressed, although there is no formal power analysis.

This paper has many interesting parallels with work on Pacific salmonid species (*Oncorhynchus*), and I encourage the authors to dig more deeply into that literature for interesting comparisons in response to stocking (and, as someone who works mostly on those species, I commit to paying more attention to papers on *Salmo* species.)

Line 62; loss of genetic diversity in itself does not lead to inbreeding, and outbreeding would increase genetic diversity. Suggest editing this sentence for clarity.

Line 78; I think it would be clearer to add the word 'only', ie '...situation where only limited introgression...'

125-126; Can you provide any information on how much hatchery stocking actually occurs in the 'wild' populations? Or are they purely natural spawning systems?

Results

Line 266; I understand what you mean by 'dominated', but it would be good to define. Occurred at >50% frequency? Or whatever criteria you want.

Line 274; Did you consider other measures of genetic diversity besides heterozygosity? For microsatellites, allelic richness is a good measure, but for SNPs it can be informative to look at the proportion of polymorphic loci. This is just a suggestion, heterozygosity works.

Line 283; "A significant divergence in average pairwise F_{st} ..." Do you mean 'decrease' instead of divergence?

Lin 309; '...corresponding fairly well...' Perhaps reword, as Figure 3 doesn't provide any indication of the correlation between estimate N_e and river size or potential population size.

320; the novelty claim 'for the first time' is inaccurate for such a global statement; it may be true for this specific system, but such a claim seems unnecessary.

364-366; For non-salmonid biologists I think you should be clearer here about the implications of these releases directly into the ocean, i.e. that it prevents the juveniles from imprinting on their natal rivers, drastically increasing straying rates.

Interpretation of relative change in large vs small populations, e.g. line 413... author attribute it to effects of high immigration, but small populations will also experience greater changes from drift, confounding this interpretation.

Line 421; should be intraspecific diversity.

Line 423; I think this needs to be tempered; if locally adapted alleles truly have a selective advantage they will not be replaced by non-adaptive alleles (unless the amount of immigration is very high). Also I think you should delete the word 'give' from this sentence.

Best regards,
Devon Pearse

Author's Response to Decision Letter for (RSPB-2020-3147.R0)

See Appendix A.

RSPB-2020-3147.R1 (Revision)

Review form: Reviewer 1

Recommendation

Accept as is

Scientific importance: Is the manuscript an original and important contribution to its field?

Excellent

General interest: Is the paper of sufficient general interest?

Excellent

Quality of the paper: Is the overall quality of the paper suitable?

Excellent

Is the length of the paper justified?

Yes

Should the paper be seen by a specialist statistical reviewer?

No

Do you have any concerns about statistical analyses in this paper? If so, please specify them explicitly in your report.

No

It is a condition of publication that authors make their supporting data, code and materials available - either as supplementary material or hosted in an external repository. Please rate, if applicable, the supporting data on the following criteria.

Is it accessible?

Yes

Is it clear?

Yes

Is it adequate?

Yes

Do you have any ethical concerns with this paper?

No

Comments to the Author

I appreciate the response to past reviewer comments. I only have one minor comment for this revision.

Line 158 mentions that mitochondrial and sex assays were developed, and three mitochondrial and five sex assays are listed in the supplemental material. I can't find mention of these markers

later in the manuscript, or anything about assigning sex to individuals. Were the mitochondrial and sex markers used for any analyses? If not, is there a reason for mentioning them in this manuscript.

Review form: Reviewer 2

Recommendation

Accept as is

Scientific importance: Is the manuscript an original and important contribution to its field?

Acceptable

General interest: Is the paper of sufficient general interest?

Acceptable

Quality of the paper: Is the overall quality of the paper suitable?

Good

Is the length of the paper justified?

Yes

Should the paper be seen by a specialist statistical reviewer?

No

Do you have any concerns about statistical analyses in this paper? If so, please specify them explicitly in your report.

No

It is a condition of publication that authors make their supporting data, code and materials available - either as supplementary material or hosted in an external repository. Please rate, if applicable, the supporting data on the following criteria.

Is it accessible?

Yes

Is it clear?

Yes

Is it adequate?

Yes

Do you have any ethical concerns with this paper?

No

Comments to the Author

I read the response to referees and checked the associated changes in the revised manuscript, but did not re-review the entire document. I find the changes in response to my previous comments and those of the other reviewer to be satisfactory, and I have not further comments to add. Nice paper.

Decision letter (RSPB-2020-3147.R1)

19-Mar-2021

Dear Dr Ostergren

I am pleased to inform you that your manuscript RSPB-2020-3147.R1 entitled "A century of genetic homogenization in Baltic salmon – evidence from archival DNA" has been accepted for publication in Proceedings B.

The referees have recommended publication, but one referee also suggests some (very) minor revision to your manuscript. Therefore, I invite you to respond to the referee's comments and revise your manuscript. Because the schedule for publication is very tight, it is a condition of publication that you submit the revised version of your manuscript within 7 days. If you do not think you will be able to meet this date please let us know.

Sincerely,
 Professor Hans Heesterbeek
 Editor, Proceedings B
<mailto:proceedingsb@royalsociety.org>

Associate Editor:
 Board Member: 1
 Comments to Author:

The authors have done an excellent job in their revisions and the response letter. Having read through both documents, I am happy to recommend acceptance without further peer review.

Reviewer(s)' Comments to Author:
 Referee: 1

Comments to the Author(s)
 I appreciate the response to past reviewer comments. I only have one minor comment for this revision.

Line 158 mentions that mitochondrial and sex assays were developed, and three mitochondrial and five sex assays are listed in the supplemental material. I can't find mention of these markers later in the manuscript, or anything about assigning sex to individuals. Were the mitochondrial and sex markers used for any analyses? If not, is there a reason for mentioning them in this manuscript.

Referee: 2

Comments to the Author(s)

I read the response to referees and checked the associated changes in the revised manuscript, but did not re-review the entire document. I find the changes in response to my previous comments and those of the other reviewer to be satisfactory, and I have not further comments to add. Nice paper.

Author's Response to Decision Letter for (RSPB-2020-3147.R1)

See Appendix B.

Decision letter (RSPB-2020-3147.R2)

24-Mar-2021

Dear Dr Ostergren

I am pleased to inform you that your manuscript entitled "A century of genetic homogenization in Baltic salmon – evidence from archival DNA" has been accepted for publication in Proceedings B.

Data Accessibility section

Open Access

Paper charges

Sincerely,
Proceedings B
mailto:proceedingsb@royalsociety.org

Appendix A

Response to Referees

Dear Editor,

Thanks for the response and comments! I am very happy to submit this revision to Proceedings. Please find below the point by point response to the reviewers' and Editors' comments (after my initials JÖ), and the adjustments made to the manuscript. A copy of the manuscript with revisions made since the previous version marked as 'tracked changes' is included in the 'response to referees' document.

The main manuscript has been submitted as a text file and figures submitted as separate files and not included within the main manuscript file. I have also followed the additional requirements for the revision made.

Sincerely,

Johan

Associate Editor

Board Member: 1

Comments to Author:

Both reviewers are very enthusiastic, and made just a few suggestions for how to improve the manuscript. Having read the paper myself, I also find the work of high quality.

Both reviewers make several constructive suggestions that the authors should be able to easily address. In particular, Reviewer 2 suggests a formal power analysis to determine whether the variation observed is sufficient for the question. Reviewer 1 asks whether the patterns observed are also evidence if you cluster the historic and contemporary samples separately.

JÖ: A formal power analysis using the program POWSIM 4.1 (Ryman and Palm, 2006) is now performed. Estimates of power to detect genetic heterogeneity for the present 82 SNP markers and average allele frequencies are presented in Supplementary material (Appendix A2). We have also added a brief description in the Material and Methods section and in the Results. Regardless of number of samples and statistical method, power was 100 % at $F_{ST}=0.015$ (and above).

A new cluster analysis with STRUCTURE is now done on historic and contemporary samples separately. Clustering was very similar as when temporal samples was analyzed together. However, for contemporary samples $K = 7$ yielded highest likelihood based on the delta K method by Evanno et al. (2005), compared to $K = 9$ for the other analyses. Results are added as a brief description (in M&M and Results) and as new figures in the Supplementary material (Figure S3).

Reference: Ryman N, Palm S (2006). POWSIM: a computer program for assessing statistical power when testing for genetic differentiation. *Molecular Ecology Notes* 6: 600-602.

Reviewer(s)' Comments to Author:

Referee: 1

Comments to the Author(s)

This study examines archival and contemporary DNA collected over 100 years to assess spatio-temporal change in Atlantic salmon populations across the Baltic sea, an area highly impacted by hydropower exploitations and associated large-scale mitigation stocking. The authors find an overall decrease in genetic divergence and diminished isolation by distance among populations. Populations in rivers with hydropower and subsequent large-scale hatchery programs showed particularly large shifts in population structure over time including changes in relative proportions of genetic clusters and reduced population differentiation. The findings of this study are relevant to ongoing conservation and commercial concerns regarding Atlantic salmon. This study will also be of interest to researchers studying Pacific salmon species, many of which face similar challenges. This study is well-designed and well-written and I only have minor comments.

Specific comments:

Line 85: “Salmon in the Baltic Sea is” should be changed to “Salmon in the Baltic Sea are”

JÖ: Text changed accordingly.

Line 90: Are the hatchery releases in Sweden or other countries?

JÖ: Text now state that the hatchery releases are in several countries, and also specify the number of released fish in Sweden.

Line 189-198: Were historic and contemporary samples examined separately with structure? It would be interesting to see if the historic samples showed even stronger clustering with contemporary samples removed from analysis.

JÖ: Yes, we have analyzed historic and contemporary samples also separately. Clustering was very similar as when temporal samples was analyzed together. However, for contemporary samples $K = 7$ yielded highest likelihood based on the delta K method by Evanno et al. (2005), compared to $K = 9$ for the other analyses. Results are added as a brief description (in M&M and Results) and as new figures in the Supplementary material (Figure S3).

Line 258: in what direction did the FIS of these sample differ? Elevated heterozygosity can be a sign of DNA contamination while very low (near zero) heterozygosity is an indicator that the wrong species was sampled. This last case happens somewhat often with pacific salmon but seems unlikely with Atlantic salmon.

JÖ: In both directions, three in negative and two in positive direction, Table S3 Supplementary material. This is now added to the text. Elevated (positive) FIS was noted in one historic and one modern sample. It is not likely that the modern sample was contaminated, since we sampled that our selves very carefully. The historic sample did not show any other signs of contamination. We also have had a very strict selection of samples to be included in the study using only samples with >95% of SNPs amplified.

Line 269-270: Can you indicate on figure 1 which populations had a significant change in the proportion of the historically dominating cluster? Maybe with an asterisk next to the cluster proportions?

JÖ: Yes, the significance is now indicated with an asterix in Figure 1.

Line 348: I like the discussion about the hatchery history on the river Luleälven. Do the other hatchery-dominated rivers have similar histories (stocking with non-native stocks) or did they use native fish to establish hatcheries? I believe this is important to note, even just briefly, either in discussion or methods because it could change the expected outcome in terms of population composition.

JÖ: Agree, the Luleälven is very interesting because of its stocking history. I have clarified in line 348 that non-native stocks were used in all hatcheries, however in limited numbers in all hatcheries/rivers but Luleälven. It is also clarified that native salmon was used as broodstock to establish all of the hatcheries.

Line 360-379: I really like the discussion on the potential for elevated straying and how this could influence the observed changes in population structure. Straying and subsequent admixture may also be consistent with the increase in expected heterozygosity (Boca, Huang, and Rosenberg 2020)

JÖ: Yes, agree. I have read the suggested literature and added the reference were appropriate.

Figure 1: It would help to have letter designations on the four top right panels, similar to the bottom four right panels.

JÖ: Letter designations are now added to the map, the four right panels as well as the bottom four right panels, and the legend is changed to explain the letter designations.

Figure 1. Are these true structure plots or summaries of the cluster proportion? I am used to seeing at least some individuals assigned to multiple clusters with structure but I don't see that here. It may just be that the size of the figures makes it difficult to see these individuals.

JÖ: The figure shows summaries of cluster proportions. We have now made this clear in the legend.

Figure 2: I like this figure. It's a nice way to show the different populations becoming more genetically similar over time.

JÖ: Thanks!

Citations:

SM Boca, L Huang, NA Rosenberg (2020) On the heterozygosity of an admixed population. *Journal of Mathematical Biology* 81: 1217-1250

Referee: 2

Comments to the Author(s)

This paper presents a population genetic analysis of modern and historical samples of Atlantic salmon from multiple populations and demonstrates changes in diversity and divergence over time. The authors attribute this change to hatchery stocking and other anthropogenic impacts. It is well done and the analysis appears sound. I found the structure of the presentation to be clear and logical. While the data consist of a small number of SNP loci by today's standards (82), they appear to provide sufficient variation for the questions addressed, although there is no formal power analysis.

This paper has many interesting parallels with work on Pacific salmonid species (*Oncorhynchus*), and I encourage the authors to dig more deeply into that literature for interesting comparisons in response to stocking (and, as someone who works mostly on those species, I commit to paying more attention to papers on *Salmo* species.)

JÖ: Yes, we broaden our scope by adding relevant literature on Pacific salmon. The response to stocking is very interesting and at the moment frequently debated on *Salmo* species. We normally also try to follow the literature on Pacific salmon since there are many similarities and important research questions relevant for both *Oncorhynchus* and *Salmo* species.

Line 62; loss of genetic diversity in itself does not lead to inbreeding, and outbreeding would increase genetic diversity. Suggest editing this sentence for clarity.

JÖ: Sentence edited for clarity as suggested.

Line 78; I think it would be clearer to add the word 'only', ie '...situation where only limited introgression...'

JÖ: The word 'only' is included as suggested.

125-126; Can you provide any information on how much hatchery stocking actually occurs in the 'wild' populations? Or are they purely natural spawning systems?

JÖ: At present, no hatchery stocking is undertaken in any of the wild populations. However, there has been historic stocking in all of the wild rivers at some point. This stocking was mainly done with locally caught salmon (i.e. supportive breeding), but included the occasional use of non-native salmon. This information is now added in the M&M section. In the Discussion, we provide information on the wild rivers where historical stocking in high numbers have been done (lines 356-364, 379-384). In figure S1 Supplemental material we show the total nr of stocking of hatchery reared salmon in the Baltic Sea in relation to the samples in the study.

Results

Line 266; I understand what you mean by ‘dominated’, but it would be good to define. Occurred at >50% frequency? Or whatever criteria you want.

JÖ: Good comment. We have now defined it as occurred at >46% frequency (which corresponded to at least twice the size of the second largest cluster). Now edited in the text.

Line 274; Did you consider other measures of genetic diversity besides heterozygosity? For microsatellites, allelic richness is a good measure, but for SNPs it can be informative to look at the proportion of polymorphic loci. This is just a suggestion, heterozygosity works.

JÖ: As the reviewer mentioned, we have looked at allelic richness, but the results were less informative compared to measure of expected heterozygosity. Also, given that the markers were initially selected based on their high variability, the use of the proportion of polymorphic loci was less informative than He

Line 283; “A significant divergence in average pairwise F_{st} ...” Do you mean ‘decrease’ instead of divergence?

JÖ: Yes, thanks. Now changed accordingly.

Lin 309; ‘...corresponding fairly well...’ Perhaps reword, as Figure 3 doesn’t provide any indication of the correlation between estimate N_e and river size or potential population size.

JÖ: We have rephrased the sentence so it is less vague. Also, a reference to the Table S1 Supplementary material is now added. In table S1, river catchment size is included. We find it relevant to give the reader information on the different river sizes, and it is also interesting (although expected) that estimated N_e is in line with the size of the rivers, i.e. large rivers larger N_e , small rivers smaller N_e .

320; the novelty claim ‘for the first time’ is inaccurate for such a global statement; it may be true for this specific system, but such a claim seems unnecessary.

JÖ: Agree, statement is now removed.

364-366; For non-salmonid biologists I think you should be clearer here about the implications of these releases directly into the ocean, i.e. that it prevents the juveniles from imprinting on their natal rivers, drastically increasing straying rates.

JÖ: Yes. A sentence clarifying the implications is now added.

Interpretation of relative change in large vs small populations, e.g. line 413... author attribute it to effects of high immigration, but small populations will also experience greater changes from drift, confounding this interpretation.

JÖ: We agree that there might be expected to be increased drift in smaller populations. However we found that in these populations that genetic changes has been most pronounced, even in the presence of such an influence. We have now made this clear in the text.

Line 421; should be intraspecific diversity.

JÖ: Changed accordingly.

Line 423; I think this needs to be tempered; if locally adapted alleles truly have a selective advantage they will not be replaced by non-adaptive alleles (unless the amount of immigration is very high). Also I think you should delete the word 'give' from this sentence.

JÖ: We have rephrased the sentence to include time perspective. If non-native fish breed with a wild stock, they will indeed introduce non-native, non-locally adapted alleles. These might indeed be removed by selection, but there is good evidence that this process can take a very long time. Also, the word 'give' is removed.

Best regards,

Devon Pearse

Journal Name: Proceedings of the Royal Society B

Journal Code: RSPB

Print ISSN: 0962-8452

Online ISSN: 1471-2954

Journal Admin Email: proceedingsb@royalsociety.org

MS Reference Number: RSPB-2020-3147

Article Status: SUBMITTED

MS Dryad ID: RSPB-2020-3147

MS Title: A century of genetic homogenization in Baltic salmon – evidence from archival DNA

MS Authors: Ostergren, Johan; Palm, Stefan; Gilbey, John; Spong, Göran; Dannewitz, Johan; Königsson, Helena; Persson, John; Vasemägi, Anti

Contact Author: Johan Ostergren

Contact Author Email: Johan.Ostergren@slu.se

Contact Author Address 1:

Contact Author Address 2:

Contact Author Address 3:

Contact Author City: Drottningholm

Contact Author State:

Contact Author Country: Sweden

Contact Author ZIP/Postal Code: 17893

Keywords: *Salmo salar*, historical DNA, human-induced, genetic change, conservation

Abstract: Intra-species genetic homogenization arising from anthropogenic impacts is a major threat for biodiversity. However, few taxa have sufficient historical material to systematically quantify long-term genetic changes. Using archival DNA collected over ~100 years, we assessed spatio-temporal genetic change in Atlantic salmon populations across the Baltic Sea, an area heavily impacted by hydropower exploitation and associated large-scale mitigation stocking. Analysis was carried out by screening 82 SNPs in 1680 individuals from 13 Swedish rivers. We found an overall decrease in genetic divergence and diminished isolation by distance among populations, strongly indicating genetic homogenization over the past century. We further observed an increase in genetic diversity within populations consistent with increased gene flow. Temporal genetic change was lower in larger wild populations than in smaller wild and hatchery-reared ones, indicating that larger populations have been able to support a high number of native spawners in relation to immigrants. Our results demonstrate that stocking practices of salmon in the Baltic Sea have led to homogenization of populations over the last century, potentially compromising their ability to adapt to environmental change. Stocking of reared fish is common worldwide and our study is a cautionary example of the potentially long-term negative effects of such activities.

EndDryadContent.

Appendix B

Response to Referees

Dear Editor,

Thanks for the response and acceptance of the manuscript! I am very pleased to submit the final version of the manuscript Proceedings B. There was a very minor suggested revision to the manuscript that I have completed. Please find below the point by point response to the reviewers' and Editors' comments (after my initials JÖ), and the adjustments made to the manuscript. A copy of the manuscript with revisions made since the previous version marked as 'tracked changes' is included in the 'response to referees' document.

The main manuscript has been submitted as a text file and figures submitted as separate files and not included within the main manuscript file. I have also followed the additional requirements for the revision made.

Sincerely,

Johan

Associate Editor

Associate Editor:

Board Member: 1

Comments to Author:

The authors have done an excellent job in their revisions and the response letter. Having read through both documents, I am happy to recommend acceptance without further peer review.

Reviewer(s)' Comments to Author:

Referee: 1

Comments to the Author(s)

I appreciate the response to past reviewer comments. I only have one minor comment for this revision.

Line 158 mentions that mitochondrial and sex assays were developed, and three mitochondrial and five sex assays are listed in the supplemental material. I can't find mention of these markers later in the manuscript, or anything about assigning sex to individuals. Were the mitochondrial and sex markers used for any analyses? If not, is there a reason for mentioning them in this manuscript.

JÖ: Agree, the mitochondrial and sex markers was not used for any analyses in the manuscript so I have removed this sentence.

Referee: 2

Comments to the Author(s)

I read the response to referees and checked the associated changes in the revised manuscript, but did not re-review the entire document. I find the changes in response to my previous comments and those of the other reviewer to be satisfactory, and I have no further comments to add. Nice paper.

JÖ: Thanks!